# Collaborative nowcasting of COVID-19 hospitalization incidences in Germany

**Daniel Wolffram**[1,2]*, **Sam Abbott**[3,4], **Matthias an der Heiden**[5], **Sebastian Funk**[3,4], **Felix Günther**[6], **Davide Hailer**[1], **Stefan Heyder**[7], **Thomas Hotz**[7], **Jan van de Kassteele**[8], **Helmut Küchenhoff**[9,10], **Sören Müller-Hansen**[11], **Diellë Syliqi**[9], **Alexander Ullrich**[5], **Maximilian Weigert**[9,10], **Melanie Schienle**[1,2], **Johannes Bracher**[1,2]

**1** Chair of Statistical Methods and Econometrics, Karlsruhe Institute of Technology (KIT), Karlsruhe, Germany, **2** Computational Statistics Group, Heidelberg Institute for Theoretical Studies (HITS), Heidelberg, Germany, **3** Department of Infectious Disease Epidemiology, London School of Hygiene & Tropical Medicine, London, United Kingdom, **4** Centre for Mathematical Modelling of Infectious Diseases, London School of Hygiene & Tropical Medicine, London, United Kingdom, **5** Robert Koch Institute, Berlin, Germany, **6** Department of Mathematics, Stockholm University, Stockholm, Sweden, **7** Institute of Mathematics, Technische Universität Ilmenau, Ilmenau, Germany, **8** Center for Infectious Disease Control, National Institute for Public Health and the Environment (RIVM), Bilthoven, the Netherlands, **9** Statistical Consulting Unit StaBLab, Department of Statistics, Ludwig Maximilian University of Munich, Munich, Germany, **10** Munich Center for Machine Learning (MCML), Munich, Germany, **11** Süddeutsche Zeitung, Munich, Germany

* daniel.wolffram@kit.edu

**Data Availability Statement:** The authors confirm that all data underlying the findings are fully available without restriction. The nowcasts collected for this study are available in a GitHub

## Abstract

Real-time surveillance is a crucial element in the response to infectious disease outbreaks. However, the interpretation of incidence data is often hampered by delays occurring at various stages of data gathering and reporting. As a result, recent values are biased downward, which obscures current trends. Statistical nowcasting techniques can be employed to correct these biases, allowing for accurate characterization of recent developments and thus enhancing situational awareness. In this paper, we present a preregistered real-time assessment of eight nowcasting approaches, applied by independent research teams to German 7-day hospitalization incidences during the COVID-19 pandemic. This indicator played an important role in the management of the outbreak in Germany and was linked to levels of non-pharmaceutical interventions via certain thresholds. Due to its definition, in which hospitalization counts are aggregated by the date of case report rather than admission, German hospitalization incidences are particularly affected by delays and can take several weeks or months to fully stabilize. For this study, all methods were applied from 22 November 2021 to 29 April 2022, with probabilistic nowcasts produced each day for the current and 28 preceding days. Nowcasts at the national, state, and age-group levels were collected in the form of quantiles in a public repository and displayed in a dashboard. Moreover, a mean and a median ensemble nowcast were generated. We find that overall, the compared methods were able to remove a large part of the biases introduced by delays. Most participating teams underestimated the importance of very long delays, though, resulting in nowcasts with a slight downward bias. The accompanying prediction intervals were also too narrow for almost all methods. Averaged over all nowcast horizons, the best performance was achieved by a model using case incidences as a covariate and taking into

repository (https://github.com/KITmetricslab/hospitalization-nowcast-hub), with a stable release published at https://zenodo.org/record/7828604. The repository also contains the truth data used for evaluation. Code to reproduce results and figures are provided at https://github.com/dwolffram/hospitalization-nowcast-hub-evaluation. A list of the participants' code repositories can be found in Section C in S1 Appendix.

**Funding:** J. Bracher acknowledges support from the Helmholtz Foundation via the project SIMCARD, while M. Schienle acknowledges support from the Helmholtz Foundation via the projects SIMCARD and COCAP. J. Bracher, D. Wolffram and M. Schienle were moreover supported by the German Federal Ministry of Education and Research (BMBF) via the project RESPINOW. J. Bracher's work has been partly funded by the Deutsche Forschungsgemeinschaft (DFG, German Research Foundation) - project number 512483310. D. Wolffram is grateful for support from the Klaus Tschira Foundation. D. Wolffram's contribution was moreover supported by the Helmholtz Association under the joint research school HIDSS4Health - Helmholtz Information and Data Science School for Health. S. Abbott and S. Funk were supported by The Wellcome Trust (210758/Z/18/Z). F. Günther was supported by NordForsk (grant no. 105572). The funders had no role in study design, data collection and analysis, decision to publish, or preparation of the manuscript.

**Competing interests:** The authors have declared that no competing interests exist.

account longer delays than the other approaches. For the most recent days, which are often considered the most relevant in practice, a mean ensemble of the submitted nowcasts performed best. We conclude by providing some lessons learned on the definition of nowcasting targets and practical challenges.

## Author summary

Current trends in epidemiological indicators are often obscured by the fact that recent values are still incomplete. This is due to reporting delays and other types of delays. Statistical nowcasting methods can be used to account for these biases and reveal yet unobserved trends, thereby improving situational awareness and supporting public health decision-making. While numerous methods exist for this purpose, little is known about their behavior in real-time settings and their relative performance. In this paper, we compared eight different nowcasting methods in an application to COVID-19 hospitalization incidences in Germany from November 2021 to April 2022. Additionally, we combined the predictions of these methods to create so-called ensemble nowcasts. Our findings indicate that while all methods yielded practically useful results, some systematic biases in nowcasts occurred and the remaining uncertainty was generally underestimated. Combined ensemble nowcasts showed promising performance relative to individual models and thus represent a promising avenue for future research.

## 1 Introduction

During infectious disease outbreaks, real-time surveillance data contributes to situational awareness and risk management, informing resource planning and control measures. However, the timely interpretation of epidemiological indicators is often hampered by the preliminary nature of real-time data. Due to reporting delays, the most recent data points are usually incomplete and subject to retrospective upward corrections. This bias can lead to incorrect conclusions about current trends. Statistical *nowcasting* methods aim to remedy this problem by predicting how strongly preliminary data points are still going to be corrected upwards, taking into account the associated uncertainty. Nowcasts thus help to uncover current trends which are not yet visible in reported numbers.

Problems of this type have been extensively researched across various disciplines; e.g., in econometrics, the gross domestic product and the inflation rate are routinely nowcasted [1]. Methods for preliminary count data as encountered in the present work originated in the actuarial sciences, where they were developed to handle insurance claims data [2]. In epidemiology, the problem of delayed reporting has been treated in diverse contexts, including the HIV pandemic [3], foodborne Escherichia coli outbreaks [4], the 2009 influenza pandemic [5] and mosquito-borne diseases like malaria [6] and dengue [7, 8]. During the COVID-19 pandemic, the problem has seen growing interest, and new approaches tailored to a variety of settings have been suggested [9–14]. There is thus an ever-growing number of methods to statistically correct reporting delays. However, two important aspects are rarely addressed in the current literature. Firstly, few studies assess the performance of methods in real-time settings. The papers we are aware of—with [14] as an exception—contain only retrospective case studies which risk smoothing over some of the difficulties occurring in real time (e.g., major data revisions, time pressure on analysts). Also, few studies include comparisons with existing methods.

While occasionally one additional model is applied for comparison [8, 11, 13], systematic comparative assessments are lacking. Our work fills this gap by examining multiple procedures in real time, thus providing a realistic picture of nowcast performance and the arising practical challenges. By bringing together several different models, our study is moreover the first able to assess the potential of combined ensemble nowcasts.

We evaluate the different nowcasting approaches in an application to German 7-day hospitalization incidences. These have played an important role in the management of the pandemic in Germany. Indeed, in November 2021, they were defined as the key indicator to determine levels of non-pharmaceutical interventions. Via a system of thresholds [15], they played an important role in the management of the pandemic, in particular in the fall and winter of 2021. Nowcasting is of particular importance for this indicator due to the way it was defined. As will be detailed in Section 2.1, the official German hospitalization numbers published by Robert Koch Institute (RKI) are aggregated by the reporting date of the associated positive test rather than the date of hospital admission. The total time span between the case report and the hospitalization report (i.e. the "delay" that has to be predicted) thus consists of two parts: the time between the report of the positive test and hospital admission and the actual reporting delay between hospitalization and the reporting thereof. This definition led to some criticism in the public discourse but was defended as a necessary compromise between timeliness and data quality by RKI [16]. Fig 1 illustrates the nowcasting task in the context of the 7-day hospitalization incidence. It shows real-time nowcasts from 1 December 2021, 1 February 2022, and 1 April 2022. Comparison with a more stable data version from 8 August 2022 shows that in these instances, the nowcasts were able to correctly reveal the actual trends, which differed sharply from the apparent declines found in the data as available at the time of nowcasting.

The present work is based on a collaborative platform, the *COVID-19 Nowcast Hub*, which we launched soon after the hospitalization incidence became the guideline value for the German pandemic policy. It served to collect and combine real-time nowcasts from several models on a daily basis. The approach builds upon the *COVID-19 Forecast Hubs*, which during the pandemic were run in the US [17], Germany and Poland [18], and later the entire European Economic Area, Switzerland and the UK [19]. These Hubs showed that combining different epidemiological models into an ensemble can produce more robust predictions, confirming results from forecasting challenges like *FluSight* on seasonal influenza [20]. We aimed for

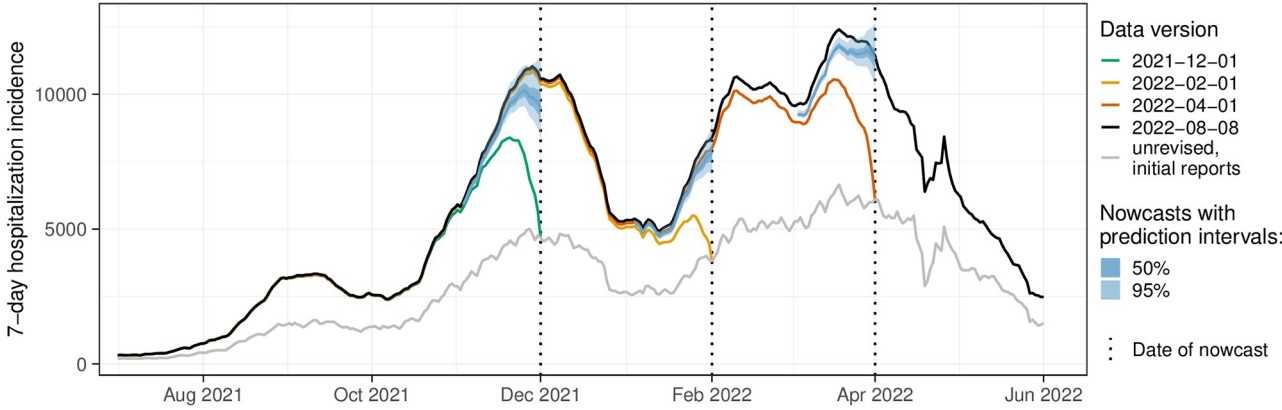

**Fig 1. Illustration of the nowcasting task.** Data available in real time (colored lines) is incomplete, and especially for recent dates, the values are considerably lower than the final corrected values (black line). Nowcasts (blue-shaded areas) aim to predict in real time what the final data points will be. The light gray line shows the initially reported value as available on the respective date.

compatibility with the Forecast Hub ecosystem in many technical and methodological aspects, in particular by following the same submission format and evaluation criteria [21]. This way we contribute to a growing evidence base on predictive epidemic modeling in real time.

The remainder of the manuscript is structured as follows. In Section 2, we introduce the 7-day hospitalization incidence as defined by RKI and outline the agreed-upon nowcast targets. We present the individual nowcasting methods and ensemble approaches, as well as the prespecified evaluation criteria. Section 3 presents the results of our formal performance evaluation, followed by qualitative observations on periods of unusual reporting patterns or the emergence of a new variant. We then assess the impact of model revisions as well as the sensitivity of the results to the exact definition of the nowcast target. Section 4 concludes with a discussion.

## 2 Methods

To facilitate a transparent assessment, we preregistered our evaluation study, specifying the criteria to assess the submitted nowcasts. The study protocol was deposited at the registry of the Open Science Foundation on 23 November 2021 [22]. In some instances, we had to deviate from the protocol. These are detailed in the respective subsections and summarized in Table A1 in S1 Appendix.

### 2.1 Definition of the COVID-19 7-day hospitalization incidence

Data on the German COVID-19 hospitalization incidence was published in a daily rhythm by Robert Koch Institute [23]. By its official definition [24], it is given by the number of hospitalized COVID-19 cases among cases reported over a 7-day period relative to 100,000 inhabitants. As illustrated in Fig 2, hospitalizations are thus aggregated by the case reporting date, more precisely, when a case was digitally registered by a local health authority, rather than the date of hospital admission (though the two may coincide). We will refer to this case reporting date as the *reference date* in the following. We note that the hospitalization is not required to occur during the 7-day window mentioned previously, nor is COVID-19 required to be the main reason for hospitalization. When new hospitalizations are added to the record, they may thus change the value of the 7-day hospitalization incidence for past periods, depending on how much time has passed between the positive test, the time of hospitalization, and ultimately

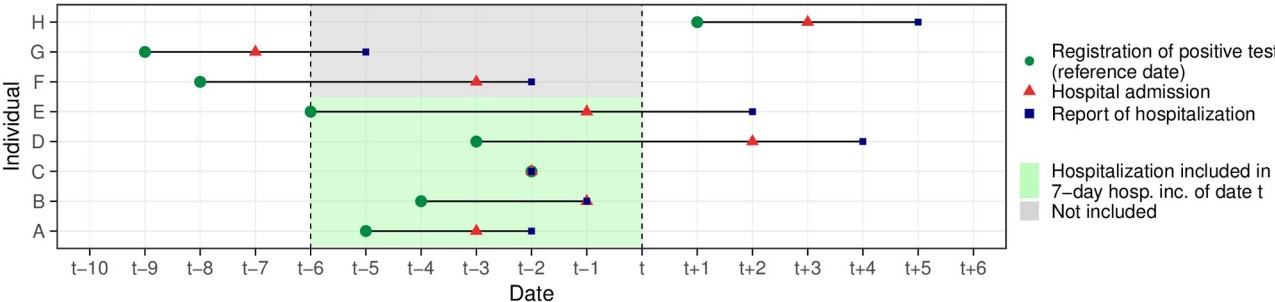

**Fig 2. Illustration of 7-day hospitalization incidences via individual-level timelines.** The reference date by which hospitalizations are counted is the date when the positive test of an ultimately hospitalized person is reported (green dots). However, hospitalizations only become known after they take place (red triangles) and are reported (blue squares). Individuals A-E are included in the 7-day hospitalization incidence of date *t* because their reference date falls within a 7-day window from *t* − 6 until *t*, even though some are reported as hospitalized later (individuals D and E). These hospitalizations only appear in the data with a delay and thus need to be predicted using a nowcasting method on day *t*. In principle, it is also possible that positive test, hospitalization and reporting all take place on the same day, as for individual C. In this case, there is no delay problem. We note that even though individuals F and G are hospitalized or reported within the period *t* − 6 to *t*, they are not counted in the 7-day hospitalization incidence for day *t* because the positive test is reported before *t* − 6. Individual H is not included because its reference date is after *t*.

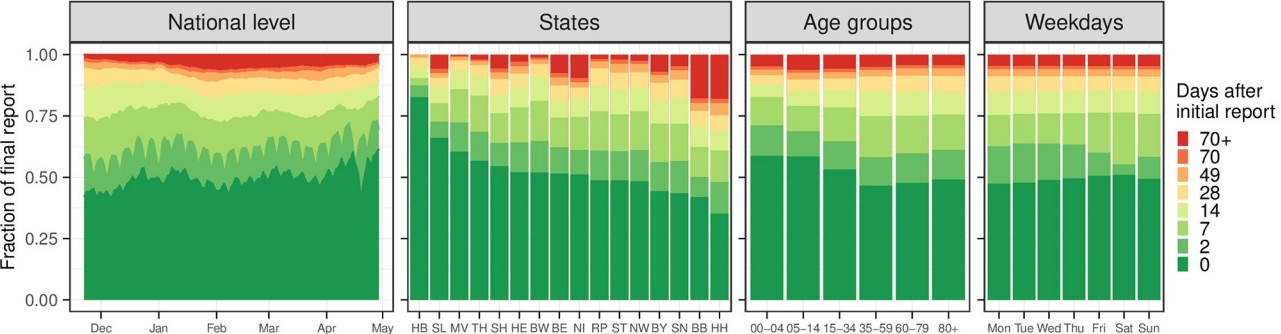

**Fig 3. Completeness of 7-day hospitalization incidences 0 to 70 days after the respective reference date.** First panel: temporal development over the considered study period, aggregated over states and age groups. Second: by state, ordered by initial reporting completeness (see Fig A1 in S1 Appendix for the definition of abbreviations). Third: by age group. Fourth: by weekday.

its reporting. Therefore, the initially reported value of the hospitalization incidence is merely an approximation and tends to be lower than the actual value.

To illustrate the extent of these revisions, Fig 3 shows the fraction of the 7-day hospitalization incidence that was reported 0–70 days after the respective reference date. Same-day values covered 50–60% of the ultimately reported hospitalizations, with a slight upward trend over the study period (left panel). Around 85% were reached after 14 days and even after 70 days, there were upward corrections of more than 3%. As illustrated in the second panel, same-day reporting completeness varied considerably across states. In Bremen (HB) it exceeded 75%, whereas it was below 50% in Saxony (SN) and Hamburg (HH). Reporting completeness was also variable across age groups and weekdays (third and fourth panels). A detailed display of temporal variations in initial reporting completeness across states can be found in Fig A1 in S1 Appendix. It should be noted that initial reporting completeness can also depend on the overall strain on the health system, and delays tend to be longer in times of high caseloads [25].

As mentioned before, thresholds of 3, 6, and 9 per 100,000 population were introduced in the fall of 2021 and used to determine the necessary extent of non-pharmaceutical interventions [15]. These were applied to the initial value of the hospitalization incidence as reported on the respective day without any retrospective completion. This value is also referred to as the *frozen value*. For illustration, these frozen values were added to Fig 1 (light gray line). We note that due to the temporal and geographic differences shown in Fig 3, the same frozen value can translate to rather different final values of the hospitalization incidence.

## 2.2 Nowcast targets and study period

The goal of the collected nowcasts was to predict how much the preliminary values of the hospitalization incidence were still going to change. Specifically, on each day during the period from Monday 22 November 2021 to Friday 29 April 2022, a prediction needed to be issued for the final value the 7-day hospitalization incidence would take for that day and the previous 28 days. In the study protocol, we defined the final state to be predicted as the time series available on 8 August 2022. This date was chosen to be 100 days after the end of our study period. Originally, teams were asked to provide nowcasts for all working days of the study period, excluding a Christmas break. However, as all teams fully automated their approaches, we were able to collect nowcasts on weekends and public holidays and include them in the study.

Teams were asked to issue nowcasts for the national level as well as for the 16 German states and seven different age groups (as available in public RKI data; 00–04, 05–14, 15–34, 35–59, 60–79 and 80+ years). No age-specific nowcasts at the state level were generated. To quantify prediction uncertainty, a probabilistic format was adopted, where teams had to submit seven quantiles (2.5%, 10%, 25%, 50%, 75%, 90%, 97.5%) of the predictive distribution in addition to the mean. Following the procedure in the various COVID-19 Forecast Hubs, our main analysis examined all outcomes on their original count scales, i.e. not standardized by population. This means that the relative size of states or age strata is reflected in the weight they receive in the overall evaluation [21].

In the study protocol, we also defined a retrospective study period reaching from 1 July 2021 to 19 November 2021. The motivation was to compare the retrospective performance on historical data available during model development to prospective performance under real-world conditions. However, due to time constraints, only two teams provided complete sets of retrospective nowcasts prior to the beginning of the prospective study. We therefore chose to omit this aspect. Instead, we added an evaluation of retrospective nowcasts from four revised models to the main study period from 22 November 2021 to 29 April 2022.

### 2.3 Overview of models

Nowcasts from eight independently run models were collected for the duration of our study. Six of them were contributed by groups of academics, one by the Robert Koch Institute (RKI) and one by the data science team of the newspaper *Süddeutsche Zeitung* (SZ). A short description of the different methods is provided in Table 1 (see also [26–32]). Most approaches took preliminary hospitalization numbers as their only input, applying various techniques to model delay distributions and the underlying time series of hospitalizations. Only the ILM model took a different approach by including the number of confirmed cases as an explanatory variable. Approaches also differed in terms of the methods used for inference, uncertainty quantification, the flexibility and complexity of their delay distribution and time series models, as well as the maximum delay considered (ranging from 35 to 84 days). Some models obtained nowcasts at a coarser spatial or age resolution by hierarchically aggregating nowcasts generated for finer strata. Models were typically not fitted to the entire available data set, but only a recent subset, the size of which again differed by team.

### 2.4 Ensemble approaches

On a daily basis, all submissions that were available at 2pm were combined to generate an ensemble nowcast, see Fig 4 for an illustration. We created the two following ensembles.

- For the MeanEnsemble each predictive quantile was obtained as the arithmetic mean of the respective quantiles of the member nowcasts. The ensemble mean was obtained as the mean of the submitted predictive means.

- For the MedianEnsemble the same procedure was applied using the median rather than the arithmetic mean for aggregation.

This direct aggregation at the level of quantiles rather than, e.g., probability density functions, is known as *Vincentization* [33]. A discussion of its properties and differences to the aggregation of density functions can be found in [34]. As the expected number of contributed models was moderate, the MeanEnsemble was expected to be better-behaved than the MedianEnsemble, which can produce oddly shaped distributions in such settings [18]. The

**Table 1. Description of contributed nowcast models.**

| Abbreviation | Short description | Reference | Generation of prediction intervals | Data input | Weekday effects | Max. delay | Length of training data | Hierarchical aggregation |
|---|---|---|---|---|---|---|---|---|
| Epiforecasts | Bayesian model assuming the underlying curve of hospitalizations follows a random walk on the log scale. Reporting delays are assumed to follow a lognormal distribution with time-varying parameters. Report date effects are handled via a random effect for day of the week. | [26] | Bayesian posterior distribution | Hospitalizations | Yes | 40 days | 40 days | No |
| ILM | Hospitalization probabilities given a positive test are estimated separately per delay time and age group. These are used to predict yet unreported hospitalizations based on case incidences. | [27] | Based on past nowcast errors | Hospitalizations, cases | Indirectly via aggregation | 84 days | 91 days | Yes |
| KIT | A simple multiplication factor approach, with uncertainty intervals generated by comparing past point nowcasts to observations. | Section E in S1 Appendix | Based on past nowcast errors | Hospitalizations | No | 40 days | 60 days | No |
| LMU | Nowcasts are based on a generalized additive model, with the delay distribution described by a sequential multinomial model. | [28, 29] | Parametric bootstrap approach using the covariance of the estimated model parameters and a Poisson observation model | Hospitalizations | Yes | 40 days | 56 days | Partially |
| RIVM | Counts per reference date and delay are modeled by a two-dimensional P-spline surface and covariates including weekday effects. This surface is extrapolated to fill in yet unknown values. | [30] | Parametric bootstrap approach using the covariance of the estimated model parameters and a negative binomial observation model | Hospitalizations | Yes | 42 days | 84 days | No |
| RKI | The conditional reporting delay probabilities are modeled using a logistic regression model that incorporates weekday, federal state, and two age groups as covariates ($\leq 60$, $> 60$). | [31, 32] | Sampling from model-based distribution | Hospitalizations | Yes | 40 days | 68 days | Yes |
| SU | Similarly to Epiforecasts, the latent curve of daily hospitalizations is assumed to follow a random walk on the log scale. The delay distribution is modeled via a discrete-time hazard model with weekday effects for the reporting day. Entries of the reporting triangle are assumed to follow a negative binomial distribution. | [9] | Bayesian posterior distribution | Hospitalizations | Yes | 35 days | 56 days | No |
| SZ | Nowcasts are based on the empirical distribution of the relative difference between initially reported and retrospectively completed values of the hospitalization incidence. | – | Empirical quantiles of past relative corrections | Hospitalizations | No | – | 60 days | No |

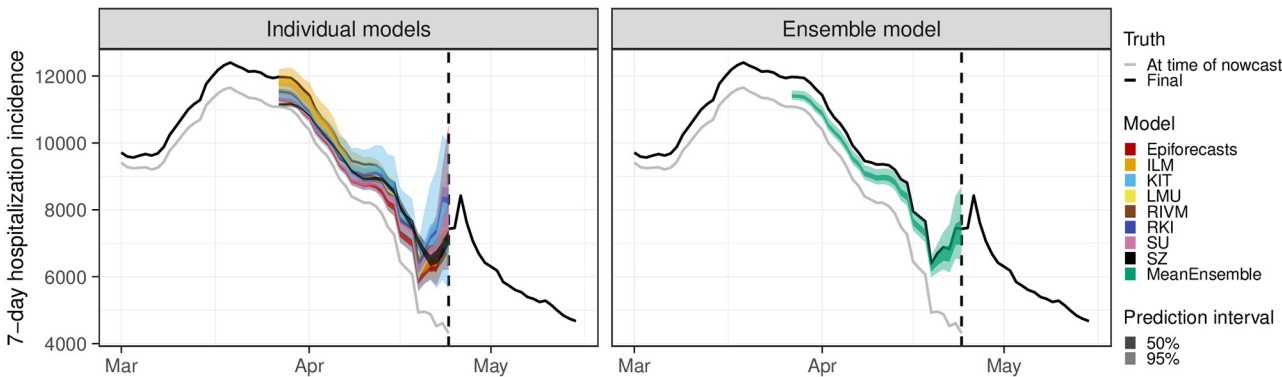

**Fig 4. Illustration of an ensemble approach.** A set of individual nowcasts can be combined into an ensemble nowcast with different aggregation approaches. Here, the ensemble is computed as the quantile-wise mean of all nowcasts.

`MeanEnsemble` was therefore prespecified as the primary ensemble approach (unlike in e.g. [17] or [18]).

We note that when using these aggregation approaches, quantile crossing can occur, meaning that the reported quantiles may not consistently increase with their nominal level [35]. To address this, one straightforward approach is to sort the quantiles in ascending order, which can improve the overall performance and coherence of the ensemble nowcasts [36]. Regrettably, this consideration was overlooked in our real-time ensemble, leading to some instances of quantile crossing. These were primarily caused by missing entries for a specific quantile level in one of the member models, as mentioned in Table A4 in S1 Appendix. As this occurred only in a small fraction of instances, we consider the impact on overall results negligible.

## 2.5 Evaluation metrics

*Proper scoring rules* are an established tool to evaluate probabilistic forecasts [37], or, in our setting, nowcasts. They are constructed such that they encourage honest forecasting, i.e., forecasters optimize their expected score by reporting their true beliefs about the future. Put differently, there is no way of "gaming" the system and obtaining improved scores by reporting modified versions of one's actual prediction. As in our setting nowcasts consist of three nested central prediction intervals, a natural choice is the interval score [37]. For an interval $[l, u]$ at the level $(1 - \alpha)$, $\alpha \in (0, 1)$, reaching from the $\frac{\alpha}{2}$- to the $\left(1 - \frac{\alpha}{2}\right)$-quantile of the predictive distribution $F$, it is defined as

$$\text{IS}_\alpha(F, y) = (u - l) \; + \; \frac{2}{\alpha} \times (l - y) \times \mathbb{1}(y < l) \; + \; \frac{2}{\alpha} \times (y - u) \times \mathbb{1}(y > u), \tag{1}$$

where $\mathbb{1}$ is the indicator function and $y$ is the realized value. Here, the first term characterizes the spread of the predictive distribution, the second penalizes overprediction (observations fall below the prediction interval) and the third term penalizes underprediction. To assess all submitted quantiles of the predictive distribution jointly we use the weighted interval score (WIS; [21]), which is a weighted average of interval scores at different nominal levels and the absolute error. For $N$ nested prediction intervals it is defined as

$$\text{WIS}(F, y) = \frac{1}{2N + 1} \times \left( |y - m| \; + \; \sum_{k=1}^{N} \alpha_k \times \text{IS}_{\alpha_k}(F, y) \right), \tag{2}$$

where $m$ is the predictive median and in our setting $N = 3$ and $\alpha_1 = 0.5$, $\alpha_2 = 0.2$, $\alpha_3 = 0.05$. We note that it is equivalent to the mean pinball loss across the respective quantile levels, which is often employed in quantile regression [21]. The WIS approximates the widely used continuous ranked probability score (CRPS) and can be interpreted as a generalization of the absolute error to probabilistic predictions. It is negatively oriented, meaning that lower values are better. The decomposition of the interval score into spread, overprediction, and underprediction also translates to the WIS and can be used to enhance the interpretability of results.

To put results into perspective, we defined the simplistic baseline model `FrozenBaseline` which applies no correction and just issues the current data version as its deterministic nowcast (i.e., with all quantiles set to the same value). This allowed us to compute relative scores

$$\text{relative WIS of model } m = \frac{\text{mean WIS achieved by model } m}{\text{mean WIS achieved by baseline model}},$$

characterizing the improvement over the uncorrected time series. Here, lower values are better, and values below 1 imply that the nowcasts reduce the error of the uncorrected time series. We note that while the study protocol specified that a baseline model was to be included, its definition was only agreed upon later. We note that the `KIT` model was originally conceived as a baseline model, but later considered too complex for this purpose; in the preregistration, it is therefore referred to as a "reference model".

To assess the central tendency of nowcasts we used the mean absolute error for predictive medians and the mean squared error for predictive means (i.e., for each functional we use the respective *consistent scoring function* [38]). To evaluate calibration, i.e., the statistical consistency between nowcasts and observations, we consider the empirical coverage of the 50% and 95% prediction intervals,

$$\text{coverage} = \frac{\# \text{ times nowcast intervals covered the final value}}{\# \text{ of nowcast intervals issued}}.$$

In case of missing submissions, i.e., if a team failed to provide a nowcast on a given day, nowcasts could be filled in retrospectively. To assess whether this had a substantial impact on the comparative evaluation, we applied a pairwise comparison scheme as described in [17] to compare models using only the sets of nowcast tasks treated in real time by each model. Details can be found in Section D in S1 Appendix.

## 3 Results

### 3.1 Completeness of submissions

All participating teams produced nowcasts over the entire study period and only rarely failed to submit nowcasts in time (see Table A2 in S1 Appendix). In most cases, missing nowcasts were filled in retrospectively. In very few cases (0.3% of all targets; see Table A3 in S1 Appendix) it was not possible to obtain submissions from all teams; to handle these cases we chose to slightly deviate from the study protocol and omit the respective targets in our evaluation. The `ILM` model did not provide state-level nowcasts, while the `RKI` model did not include age-stratified results. Moreover, the `RKI` model only provided point nowcasts and two quantiles in real time (at levels 2.5% and 97.5%); the remaining quantiles were only provided in retrospect. We encountered some more minor difficulties, e.g., due to missing quantiles for certain targets; we summarize these and the chosen solutions in Table A4 in S1 Appendix.

### 3.2 Visual inspection of nowcasts

For a first impression of nowcast performance, Fig 5 shows same-day nowcasts at the national level (i.e., at each date the respective nowcast with a horizon of 0 days is shown). Fig 6 shows the same for nowcasts 14 days back in time (i.e., for each day the nowcast issued 14 days later is shown). Displayed are the median predictions along with the central 50% and 95% prediction intervals. The light gray line shows the data as available when the nowcast was issued (which in Fig 5 corresponds to the *frozen values*), and the red line shows the respective final value as available on 8 August 2022. In both figures, it can be seen that nowcasts from all models are generally close to the final values to be predicted. However, considerable variability in interval widths is apparent, ranging from rather wide (KIT) to very narrow intervals (LMU, RKI). Some models, in particular KIT and SZ, display pronounced weekday patterns in their same-day nowcasts, which to a lower degree also carry through to the ensemble nowcasts. For the nowcasts 14 days back in time we observe a slight downward bias in the central tendency, the only exception being the ILM model. As most of the concerned models moreover feature quite narrow prediction intervals, these often do not cover the final values.

### 3.3 Formal evaluation

To consolidate the qualitative findings from the previous section, we turn to a formal evaluation and consider evaluation scores and interval coverage rates. Fig 7 displays the mean and relative WIS values achieved by different models for the three considered aggregation levels (national level, states, and age groups). The left column shows mean scores (on the absolute

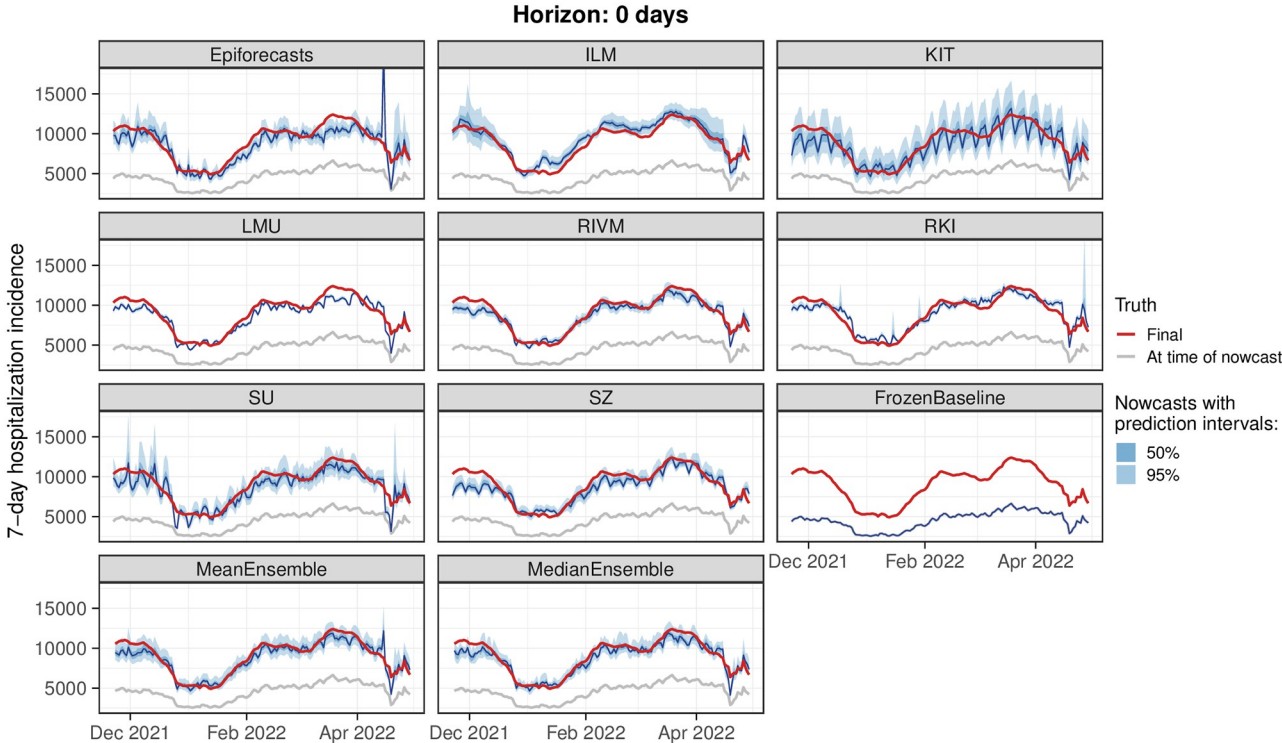

**Fig 5. Nowcasts with a horizon of 0 days back.** Same-day nowcasts of the 7-day hospitalization incidence as issued on each day of the study period. Nowcasts are shown for the German national level.

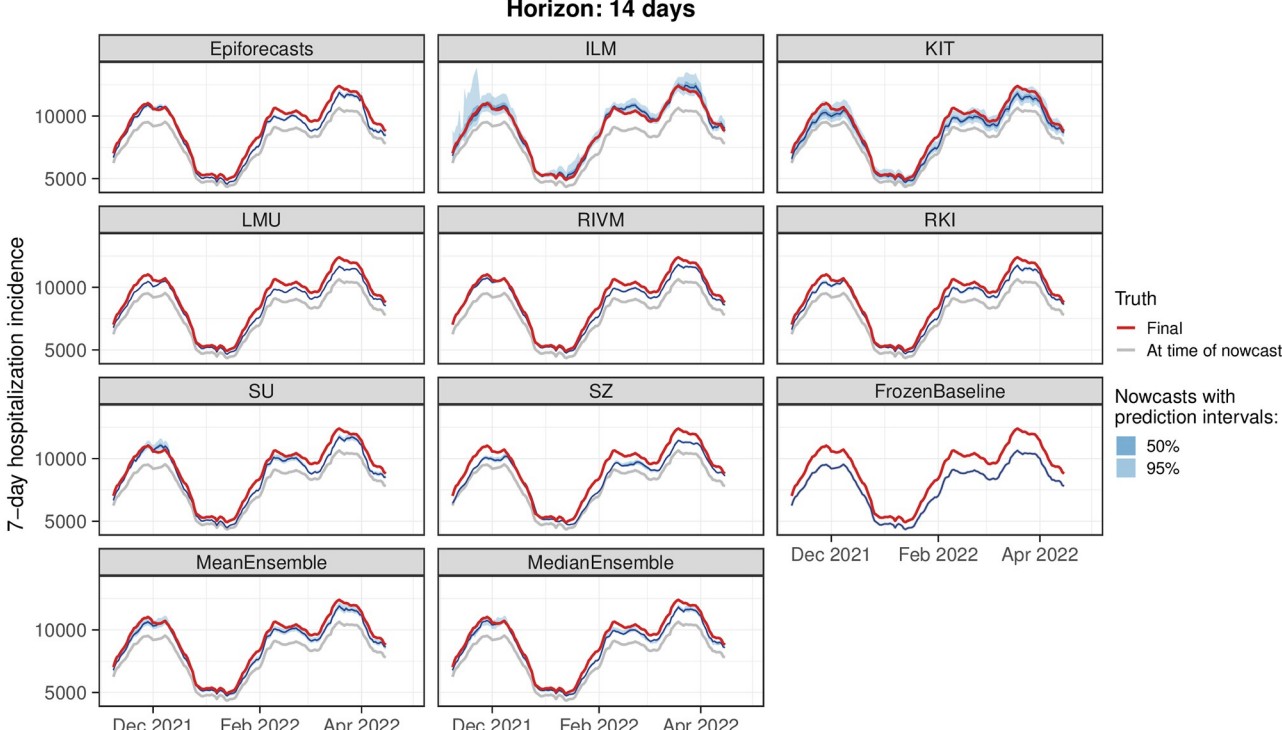

**Fig 6. Nowcasts with a horizon of 14 days back.** Nowcasts of the 7-day hospitalization incidence as issued 14 days after the respective date. Nowcasts are shown for the German national level.

and relative scale) across all strata and horizons, decomposed into contributions of spread, underprediction, and overprediction. The middle and right panels show the mean WIS and relative WIS by horizon, respectively (-28 to 0 days; see Section 2.5). At the national level and across age groups, the overall scores of the `ILM` model were considerably lower than the scores of all other models. The stratification by horizon indicates that it performed especially well for nowcasts seven or more days back. For the most recent days (-3 to 0 at the national level, -6 to 0 for age groups) the `MeanEnsemble` performed best. Across states, the `MeanEnsemble` outperformed the other models for horizons of -11 to 0 days. For horizons of -28 to -12 days, the `KIT` model achieved the best scores, which (by a narrow margin) led to the best overall result pooled across horizons. The relative scores indicate that, pooled over all horizons, most models were able to reduce the error of the uncorrected time series (`FrozenBaseline`) by roughly 80% (relative WIS of 0.2), while the `ILM` model achieved a reduction of about 90% (relative WIS 0.1). It is notable that `ILM` achieved almost constant improvements across horizons, while the improvements achieved by the other models were quite modest for horizons further into the past. To allow for a more detailed exploration of results we provide a display of the distribution of model ranks across individual nowcasting tasks (Fig A7 in S1 Appendix) and of scores over time (Fig A11 in S1 Appendix). Similarly to [17], we find that the `MeanEnsemble` reliably achieved above-average performance across all locations and age groups (almost never ranking in the bottom). Additionally, stratified results by day of the week are shown in Fig A8 in S1 Appendix. For `KIT` and `SZ`, which did not account for weekday effects, we indeed observe performance differences for different weekdays. Fig A9 in S1 Appendix further shows nowcasts by `KIT` as issued on different weekdays, with a tendency for

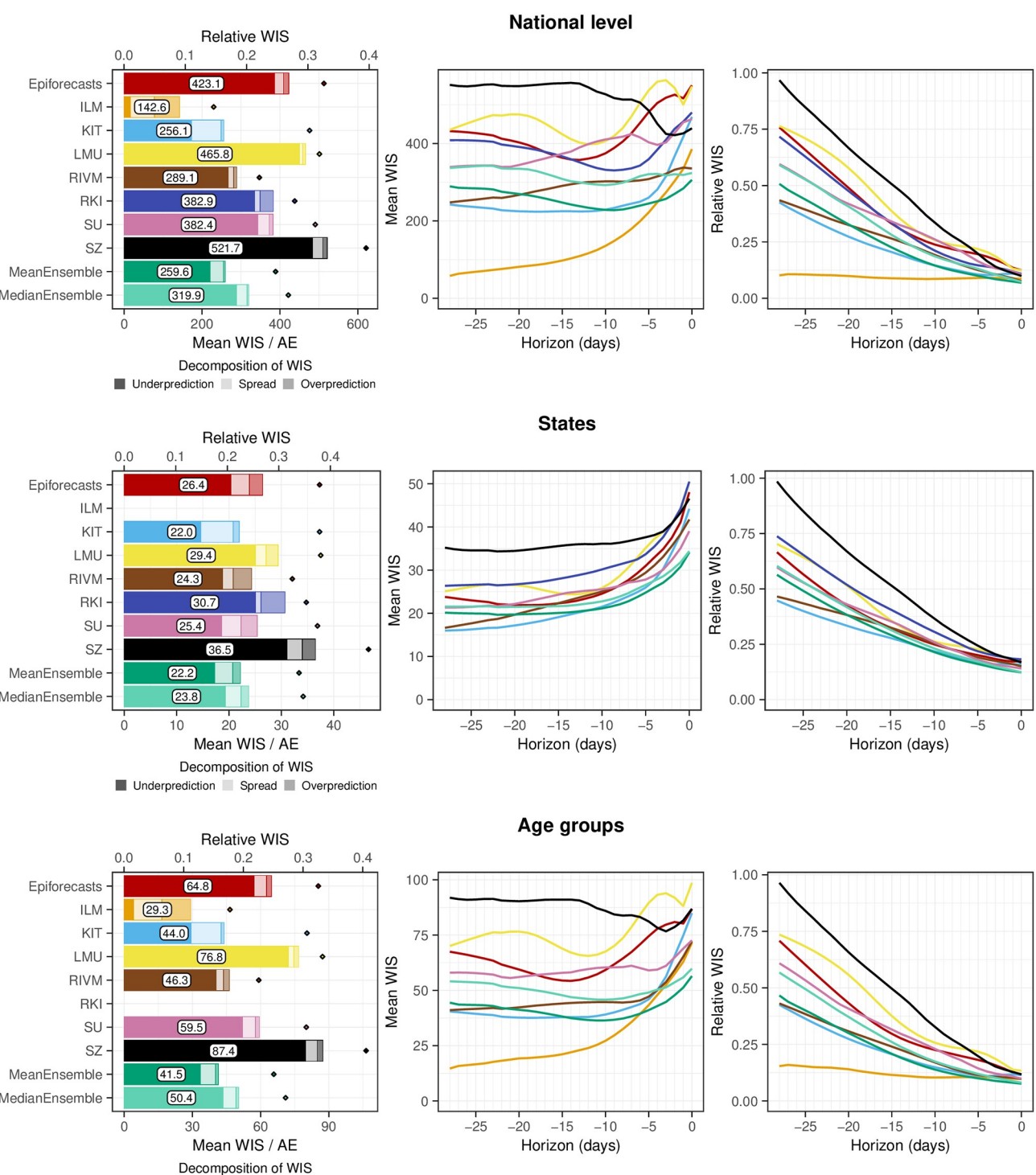

**Fig 7. Score-based performance.** Shown is the mean WIS for the national level (top) and averaged across states (middle) and age groups (bottom). The first panel in each row displays the average across all horizons (on the absolute and relative scales). The decomposition into nowcast spread, underprediction, and overprediction (see Section 2.5) is represented by blocks of different color intensities. The absolute error is indicated by a diamond (◇). The second and third panels in each row show the mean WIS and the relative WIS, respectively, stratified by horizon.

underprediction on Mondays and many cases of overprediction on Fridays or Saturdays. As seen in Fig A10 in S1 Appendix, `ILM` on the other hand did not exhibit any such differences by weekday.

Figs A4 and A5 in S1 Appendix summarize results in terms of mean absolute errors of predictive medians and mean squared errors of predictive means in the same format as in Fig 7. The `ILM` model again performed favorably. Among the remaining models, `RIVM` shows good performance, in many cases outperforming the ensembles. The `KIT` model, on the other hand, which performed relatively well on WIS, achieved below-average results.

Empirical coverage rates of the 50% and 95% prediction intervals are displayed in Fig 8. Results are stratified by aggregation level (national, states, age groups) and nowcast horizon (-28 days to 0 days). The best calibration was achieved by the `ILM` model, with coverage rates close to the nominal levels at most horizons. Only for short horizons of -10 to 0 days coverage dropped moderately. In contrast, the `KIT` model achieved higher coverage rates for horizons between -4 and 0 days, which considerably dropped for nowcasts further into the past. All other models were overconfident and did not reach the respective nominal coverage levels. As for `KIT`, coverage was lower for nowcasts further back in time, for some models to a point where only a few observations were covered at -28 days.

To assess the impact of retrospective fill-in nowcasts for missing submissions, we recomputed relative WIS values using only real-time submissions and the pairwise comparison scheme from [17]. As can be seen from Table A5 in S1 Appendix the results barely change, indicating that fill-in nowcasts did not have a relevant impact on overall scores. Furthermore, as designated in the study protocol, we reran the evaluation using hospitalization incidences per 100,000 population rather than absolute hospitalization counts. The results are displayed in Fig A12 in S1 Appendix and do not differ qualitatively from those in Fig 7.

As we consider the nowcasts for the most recent days the most relevant from a public health perspective, we conclude with an additional non-preregistered summary of scores across horizons -7 to 0 days. Fig 9 shows the average weighted interval scores and interval coverage rates. For this subset of nowcasting tasks, the `MeanEnsemble` outperformed the individual models in all three categories, closely followed by `ILM`. The `KIT` model reaches close to nominal coverage, while the other models are again overconfident.

## 3.4 Interpretation of evaluation results

As some of the presented results may seem contradictory at first sight, we provide some additional interpretations. Firstly, the opposing trends in absolute and relative WIS across horizons in Fig 7 can be interpreted as follows. All nowcasts—including the `FrozenBaseline`—get closer to the later observed final value as time passes and more complete data accumulates; thus, the absolute WIS decreases. However, most models seemed to have more difficulties predicting the small number of late additions than the bulk of early additions, leading to higher relative WIS. A possible explanation is that modelers needed to make a choice on which maximum delay to take into account. In light of Fig 3, the values of around 40 days as chosen by most teams may have been too low and led models to ignore a non-negligible fraction of hospitalizations still to be added. As can be seen from Fig 5, the resulting bias got largely absorbed in the overall uncertainty for same-day nowcasts. For the horizon of -14 days (Fig 6), on the other hand, it caused a visible shift between nowcasts and final values, which likewise led to insufficient coverage of prediction intervals.

The maximum delay chosen may also explain why the `ILM` model, which used a value of 80 rather than 40 days, was the best-performing individual method. However, the model also differed from the others in its general approach, using a regression on case incidences in addition

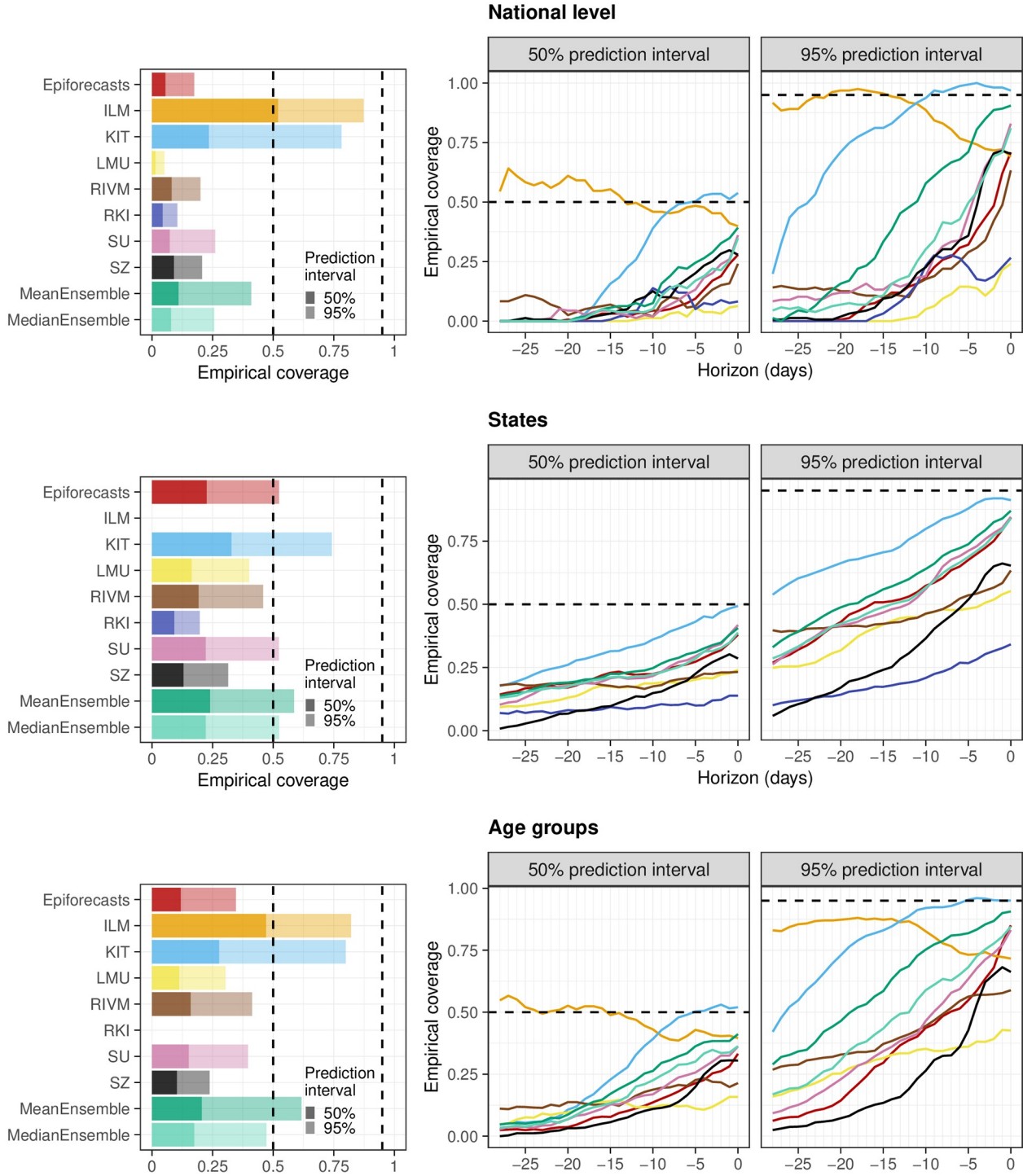

**Fig 8. Empirical coverage of the prediction intervals.** Shown is the coverage for the national level (top), across states (middle), and across age groups (bottom). The first panel in each row displays the overall coverage of the 50% and 95% prediction intervals across all horizons. The second and third panels in each row show the empirical coverage of the 50% and 95% prediction intervals, respectively, stratified by horizon. The dashed lines indicate the desired nominal levels.

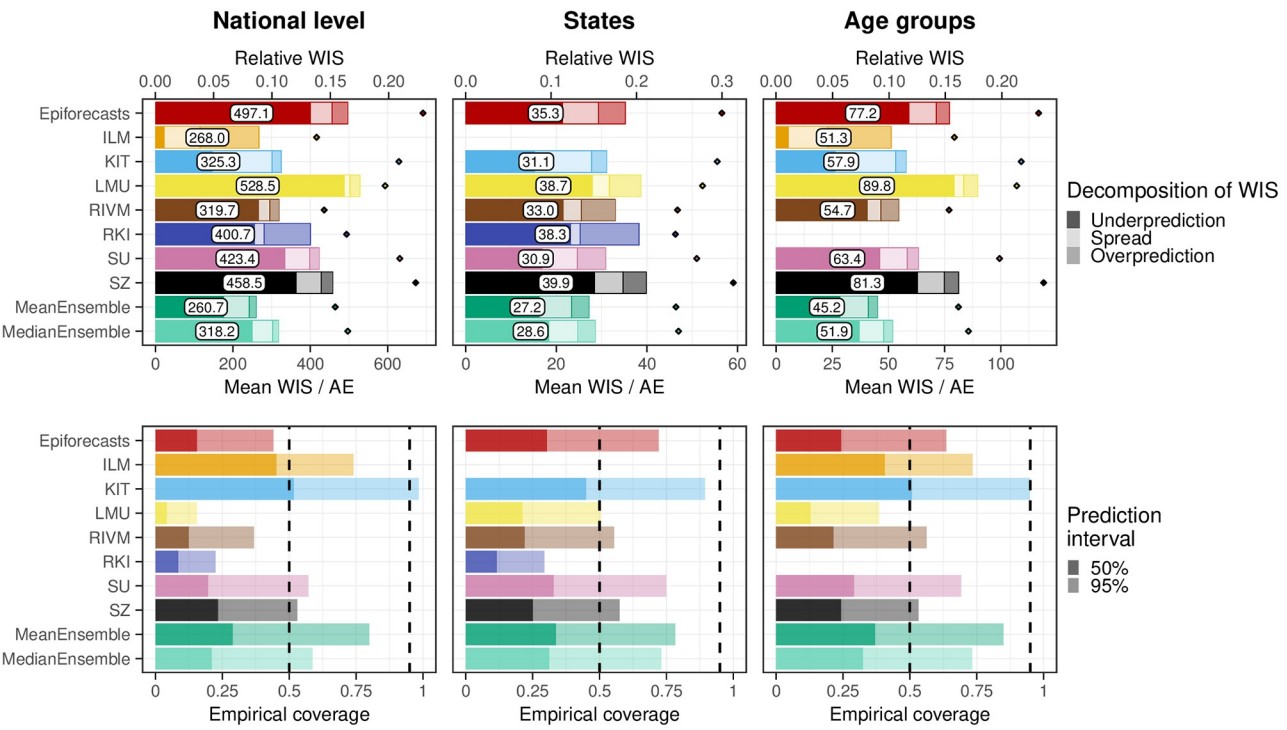

**Fig 9. Scores and coverage on short horizons.** Shown are the mean WIS with absolute errors (top) and the empirical coverage (bottom) across horizons from 0 to -7 days.

to preliminary hospitalization numbers. We will attempt to shed more light on this aspect in Section 3.6. As a last relevant difference to most other models, the ILM approach based uncertainty intervals directly on the errors of past real-time nowcasts, an approach close to the idea of conformal prediction [39]. A similar approach was also taken by the KIT model (see Section E in S1 Appendix). The fact that these two models achieved the best calibration indicates that this approach may quantify nowcast uncertainty more realistically than standard model-based uncertainty intervals.

The decomposition of the WIS into components for spread, overprediction, and underprediction [21], displayed in the left column of Fig 7, is informative on the challenges the different approaches faced. Penalties for underprediction make up a very large part of the overall scores for all models except for ILM. This confirms the observation of a downward bias from Fig 6.

The best-performing individual models ILM and KIT issued predictive distributions with higher variability than the other models, indicated by the larger spread component. As can be seen from the diamond symbols in Fig 7 and in more detail from Fig A4 in S1 Appendix, the KIT model did not issue particularly accurate point predictions (predictive medians). A likely reason is the lack of weekday effects, see Figs A8 and A9 in S1 Appendix. Its lower WIS values were primarily a result of better uncertainty quantification.

### 3.5 Impact of unusual reporting patterns and changes in virus properties

The nowcasting models in our study assumed either that the probability of hospitalization given a positive test remains roughly constant (the ILM model) or that the delay distribution in hospitalizations does so (all other models). In Fig 10, we therefore show four examples

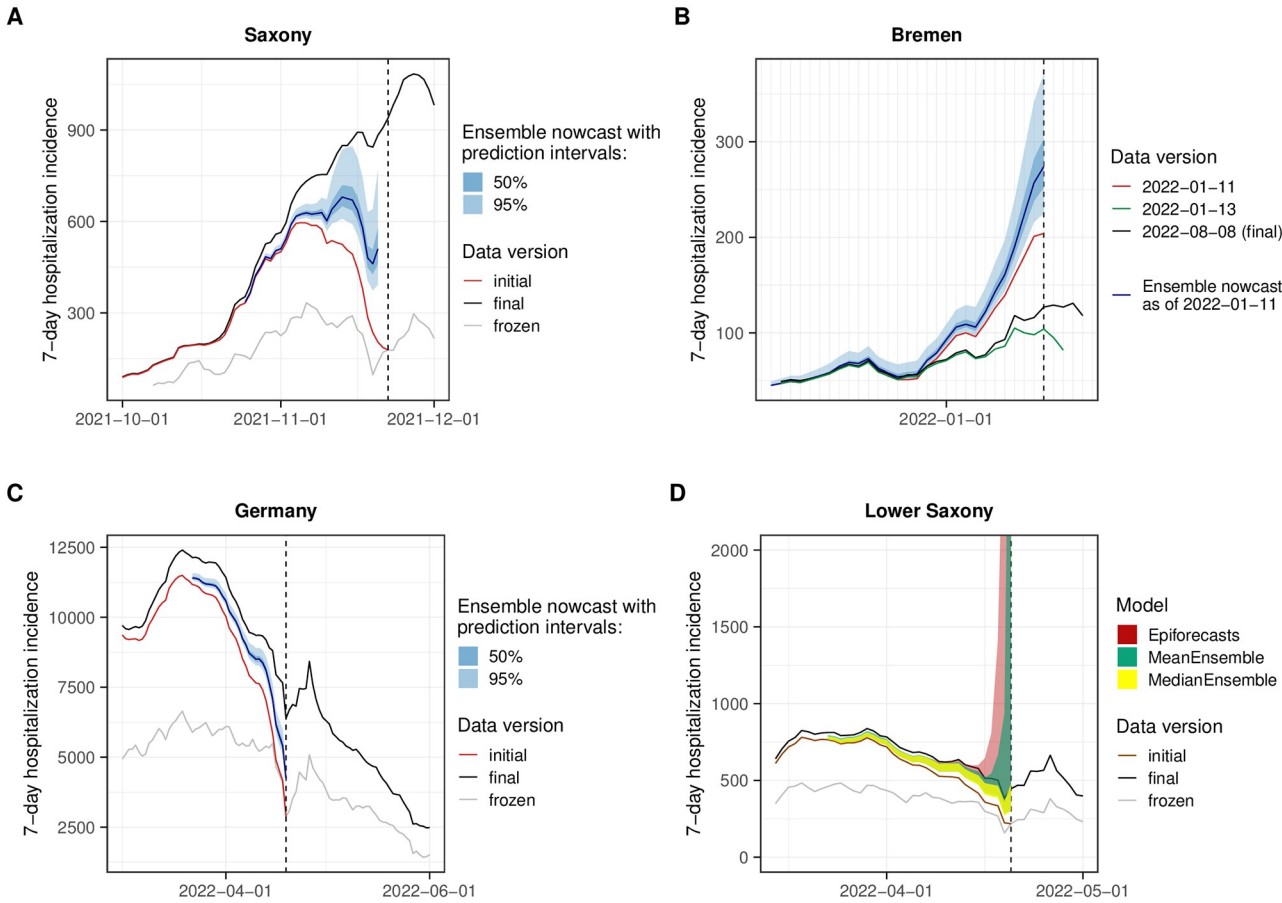

**Fig 10. Examples of time points when delay distributions were subject to sudden changes. (A)** Saxony, nowcast made on 22 November 2021: overwhelmed hospitals lead to severe underreporting and thus too low nowcasts. **(B)** Bremen, nowcast made on 11 January 2022: some incorrect entries got removed from the records, resulting in a downward correction and thus too high nowcasts. **(C)** Germany, nowcast made on 19 April 2022: following the Easter weekend with lower than usual initial reporting coverage, nowcasts were considerably too low. **(D)** Lower Saxony, nowcasts made on 20 April: after the Easter weekend, Epiforecasts issued very wide nowcast intervals, presumably due to numerical problems. The dashed lines indicate the time when the nowcasts were made. (Incidentally, in example (A), the horizons of 0 days and 1 day back were missing, see Table A3 in S1 Appendix).

where these assumptions were violated. In mid-November 2021, hospitals in Saxony were overwhelmed [40], leading to disruptions in the reporting system. As a consequence, initial reporting completeness dropped rather suddenly. This led the majority of models to underpredict, leading to an ensemble nowcast that was too low, as illustrated in Fig 10A. We note that in this instance we were aware that nowcasts for Saxony were likely unreliable and issued a warning on our website. Fig 10B shows an unusual reporting pattern from the state of Bremen from early 2022. Here, a relevant number of reported hospitalizations got removed from the record on 12 and 13 January, presumably due to faulty initial reporting. Nowcasts issued up to 11 January were thus considerably above the final data version from 8 August. Fig 10C and 10D show issues arising after the Easter weekend of 2022, when initial reporting was considerably lower than usual. As can be seen in Fig 10C, this led to too low ensemble nowcasts on Tuesday, 19 April. Also, over the following days, it seems to have caused issues in the fitting of certain models. As an example, Fig 10D shows the Epiforecasts output for Lower Saxony on 20 April. It features an excessively wide prediction interval, likely as a reaction to rapidly

shifting delay distributions in the previous days. The `MeanEnsemble`, shown in green, was strongly affected by this unusual behavior of a member nowcast, while the more robust `MedianEnsemble` remained unaffected.

A last noteworthy particularity is the behavior of the `ILM` model in January 2022, following the transition from the Delta to the Omicron variant. The Omicron variant is known to have lower clinical severity than the Delta variant [41], meaning that during the transition the ratio of hospitalizations and confirmed cases gradually declined. For the `ILM` model, which assumes this ratio to remain constant, this led to an upward bias in nowcasts, which can be discerned from Fig 5 as an upward bump not present in the other models.

### 3.6 Retrospective variations of models

We next aim to shed some more light on how different modeling choices impact performance and how learnings from our study period facilitated the improvement of methods. To this end, we assess the performance of four variations of previously discussed models, which were applied retrospectively:

- The `LMU` team implemented a new approach to generate uncertainty intervals, which like the `ILM` and `KIT` methods is based on past nowcast errors.

- The `RKI` team obtained nowcasts by aggregating over finer strata and originally assumed independence across strata to generate prediction intervals at the aggregate level. This was changed to an assumption of strong correlations across strata, leading to wider prediction intervals.

- The `KIT` team reran its model with an increased maximum delay of 80 days, in contrast to the 40 days used in real time. This also required an increased length of training data, which was set to 100 days.

- Conversely, the `ILM` model was rerun with a maximum delay of 42 rather than 84 days, which is comparable to the maximum delays used by the remaining models. This was not meant as an improvement but as an adjustment to assess the impact of longer/shorter maximum delays.

The results are shown in Fig 11. They indicate improvements across all aggregation levels and horizons for the revised `LMU`, `RKI`, and `KIT` models. In particular across age groups, the `KIT` model now came close to the performance the `ILM` model achieved in real time. The coverage proportion of prediction intervals was increased for all three models, with the updated `KIT` model even leaning towards over-coverage (too wide intervals). The `LMU` and `RKI` models, on the other hand, remained overconfident. Decreasing the maximum delay in the `ILM` model slightly reduced the overall performance on the national level, while average scores across age groups remained almost unchanged. The score decomposition shows that the adjusted model tended to underpredict (similarly to the other models), while the original model tended to overpredict. Possible explanations will be discussed in Section 4.

### 3.7 Sensitivity of results to definition of final data

In our study protocol, we specified that the final state of the time series to be predicted was the version available on 8 August 2022, i.e., 100 days after the end of the study period. However, as we became aware of the fact that data revisions could occur with considerably longer delays than initially expected, we performed a sensitivity analysis to assess the impact of this choice. Fig 12 shows how the average WIS aggregated over horizons and different levels of stratification (i.e., the results shown in the left column of Fig 7) change when using a different data

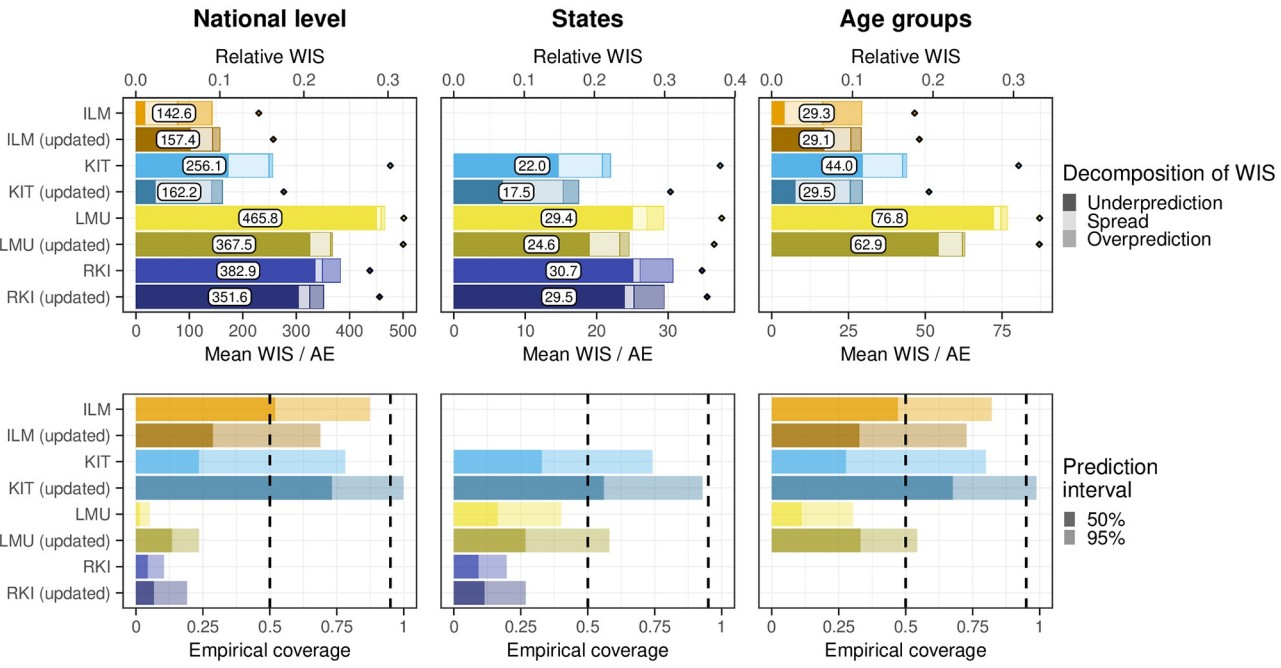

**Fig 11. Evaluation of retrospective model variations.** Comparison of variations of the `ILM`, `KIT`, `LMU`, and `RKI` models and the same models as submitted in real time. Shown are the mean WIS with absolute errors (top) and the empirical coverage (bottom). Results are comparable to those from Figs 7 and 8.

version as the final one. It can be seen that the average scores of all models, except for `ILM`, increase in parallel as newer data versions are used. The increase for `KIT` is slightly more gradual. This is because these models tend to underpredict, and as time passes and more additions are made to the data, this problem is exacerbated. For the `ILM` model, which tends to overpredict, average scores initially decline and then plateau, leading to an even more pronounced lead relative to the other models. As can be seen from Fig A6 in S1 Appendix, using a later data version for evaluation, `ILM` ultimately also surpasses the ensemble when restricting results to horizons 0 to 7 days back.

The original target allowed revisions until the final date of 8 August 2022, meaning that for different reference dates, adjustments could be made over time periods of different lengths (e.g., the hospitalization incidence for the first reference date in our study period could be completed over a longer time than that of the last reference date). Denoting the hospitalizations for reference date $t$ reported with a delay of $d$ days by $x_{t,d}$, the nowcasting target $y_t$ can formally be written as

$$y_t = \sum_{i=0}^{6} \sum_{d=0}^{t_{\max}-(t-i)} x_{t-i,d} \quad \text{with } t_{\max} = \text{2022-08-08},$$

where the first sum represents a 7-day window ending in date $t$ and the second sum accumulates the hospitalizations with reference date $t - i$ reported until $t_{max}$.

An alternative to choosing one specific data version as the nowcast target is a "rolling" approach that considers delayed reports for each reference date $t$ up to a specified maximum delay $D$. The nowcast target $z_t$ then becomes

$$z_t = \sum_{i=0}^{6} \sum_{d=0}^{D} x_{t-i,d}.$$

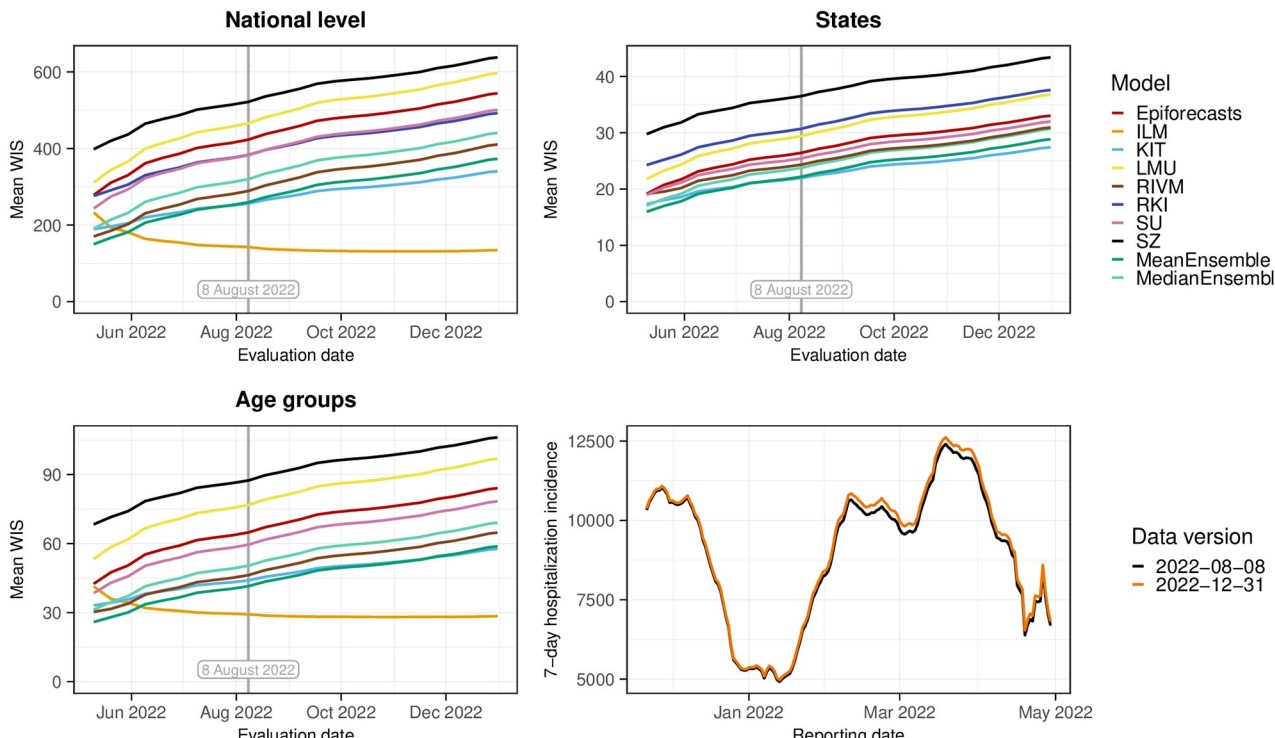

**Fig 12. Sensitivity of the scores to the chosen "final" data.** Shown is the mean WIS computed with different data versions as the target. The version prespecified in the study protocol is 8 August 2022, marked by a vertical line. Top left: national level. Top right: averaged over states. Bottom left: averaged over age groups. The bottom right panel overlays the national-level data as of 8 August and 31 December to illustrate the importance of late revisions.

In this case, the data for each reference date has the same amount of time to be revised. See Fig A3 in S1 Appendix for an illustration of these differently defined target time series. Fig 13 shows the results this approach yields for a maximum delay of 40 days, which corresponds roughly to the maximum delay used by most teams. As this target definition is better aligned with the practical implementations teams chose, it is not a surprise that the mean WIS values are lower and coverage is higher. The `ILM` model (in its adjusted version with a maximum delay of 42 days) now shows quite similar performance to the other approaches, with a tendency to overpredict. The score components for the other models are more balanced and the ensemble nowcasts clearly lead the field. Retrospectively, we think that this definition of targets might have been a more coherent and operationally meaningful approach, see Section 4 for a discussion.

## 4 Discussion

In this paper, we presented results from a preregistered study to evaluate probabilistic real-time nowcasts of the 7-day hospitalization incidence in Germany from November 2021 to April 2022. We found that all models were able to correct for a large part of the biases caused by reporting delays. Further, we identified calibration of uncertainty intervals as a major challenge, as the empirical coverage rates achieved by most models were considerably below the respective nominal levels. Reasons for insufficient coverage likely include too inflexible modeling of dispersion and delay distributions, and also the fact that most teams truncated delay distributions at a too short maximum delay. The exception was the `ILM` model which also stood

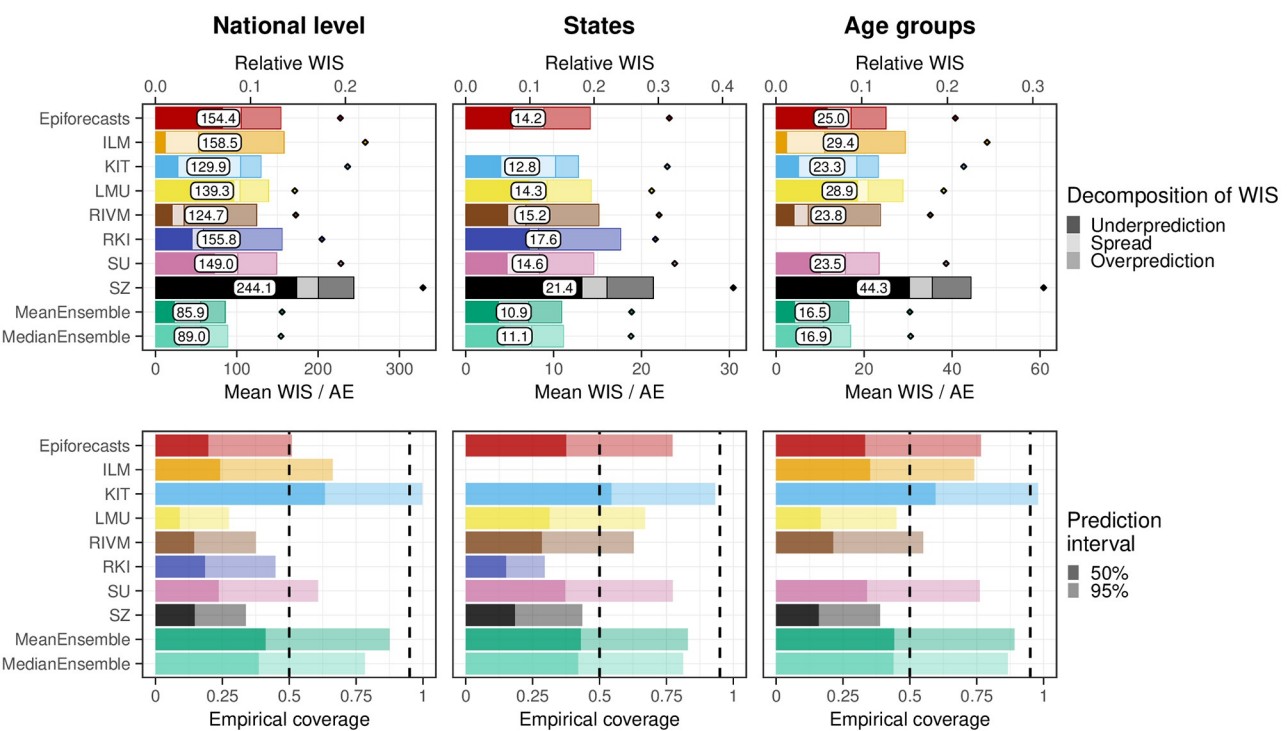

**Fig 13. Performance based on the alternative target with a maximum delay of 40 days.** Shown are the mean WIS with absolute errors (top) and the empirical coverage (bottom) computed with respect to a revised target defined as the number of hospitalizations reported with a delay of up to 40 days. For the `ILM` model we used the revised model with a maximum delay of 42 days and also recomputed the ensembles with these revised nowcasts. For the other models, the assumed maximum delays are approximately aligned with the redefined target, see Table 1.

out in terms of score-based performance for the national level and across age groups, and to a lesser degree the `KIT` model. These two models incorporated uncertainty quantification using past real-time nowcast errors which proved advantageous when compared to the standard model-based uncertainty intervals.

Our analyses from Sections 3.6 and 3.7 suggest that the success of the `ILM` model arose from the interplay of two aspects. On the one hand, it used a maximum delay that is longer than those of the other models, but, judging by Fig 3, still somewhat too short. On the other hand, the model appears to have a tendency to slightly overpredict the number of hospitalizations added up to a given maximum delay. As these two aspects work in different directions, the resulting nowcasts are overall well-aligned with the defined target (data version from 8 August 2022). Whether nowcasts taking into account case incidence inherently perform better needs to be explored in future work. A combination of different data streams may reduce the dependence of nowcasts on certain assumptions, such as the constant completeness of initial reports. The `Epiforecasts` and `SU` models have been extended in this direction. In an application to COVID-19 deaths in Sweden [42], the inclusion of reported cases and intensive care admissions as leading indicators indeed led to improved predictions.

The `MeanEnsemble`, along with the `KIT` model, performed best across federal states (for which the `ILM` model did not provide nowcasts). Also, it showed very good relative performance for horizons -7 to 0 days, as well as for the revised "rolling" target. We therefore conclude that ensemble approaches are a promising avenue in order to improve disease nowcasting. However, our case study also illustrates a limitation of unweighted ensembles.

The ensemble may have been imbalanced in the sense that a majority of its members followed similar strategies and had similar weaknesses (specifically a downward bias due to neglecting very long delays). A weighted ensemble could have capitalized on the strengths of the `ILM` model, which followed a conceptually different approach and could have served as a counterbalance. However, it is not obvious how ensemble members can be assigned weights in real time in a nowcasting setting. This represents an interesting future research area.

A difficulty we encountered in terms of our study design is that results depend on which data version is used as the "final" one (i.e., the values against which nowcasts are evaluated; Section 3.7). As the choice of 8 August 2022 was preregistered and known to all participating teams, the prediction task was well-defined, and we stuck to this choice for our main analysis. Nonetheless, this definition, which was based on the assumption that data would be stable after 100 days, turned out not to be ideal in retrospect. In particular, it implies that for the first day of our study period (22 November 2021), retrospective additions could accumulate over 259 days, while for the last (29 April 2022) this was restricted to 100 days. Defining the nowcast target in a "rolling" fashion as explored in Section 3.7 might have been a more appropriate choice. This would have been a more clearly defined modeling task, and modelers would not have had to choose a maximum delay for their models themselves.

The question of whether additions should be ignored from a certain maximum delay onward is closely linked to what these additions actually mean and whether they are relevant from a public health perspective. As mentioned in Section 2.1, the 7-day hospitalization incidence also contains hospitalizations that are not primarily due to COVID-19. These hospitalizations have been found to represent a considerable fraction [43]. It seems plausible that very long delays are due to large time differences between the positive test and hospital admission, in which case the share of hospitalizations that are not primarily due to COVID-19 may be high. Also, it can be questioned whether hospitalizations a long time after a positive test are relevant for the real-time assessment of healthcare burden. Both aspects strengthen the case for limiting nowcasting to hospitalizations reported up to a carefully chosen maximum delay.

The definition of the *frozen* values used by the Robert Koch Institute when applying legally defined thresholds can be seen as a strong form of discarding delayed hospitalizations. It has the advantage of simplicity and unambiguity, which are required for actionable guidelines in a legal context. After all, it seems difficult to integrate complex statistical methods with many tuning parameters into a binding legal document. An important downside, however, is that the same *frozen* value can mean rather different things at different time points and in different locations, due to differences in initial reporting completeness. We thus argue that outside of purely legal considerations, nowcasts can provide a more thorough picture of current developments.

All nowcasts generated within the presented collaborative project are available in a public repository (see data availability statement). Time-stamped versions of hospitalization data as available at different points in time can be retrieved from the commit history of the repository as well as directly from Robert Koch Institute [23]. We hope that this data can be of use as a benchmarking system for future nowcasting methods. In this context, we note, however, that the present paper is a comparison of *nowcasting systems*, which are given by a statistical model, but also various additional analytical choices, in particular the assumed maximum delay and the length of training data used at each time point. These decisions can have a substantial impact on predictive performance (see Section 3.6) and are easier to get right in hindsight than in real time. To ensure a fair comparison, it may therefore be reasonable to use the "rolling" target as discussed in Section 3.7.

The nowcasts produced for this project were routinely displayed by numerous German-speaking media, including *Die Zeit*, *Neue Zürcher Zeitung* and *Norddeutscher Rundfunk*.

While some displays were limited to the point nowcasts (predictive medians), others made the predictive uncertainty clearly visible. This development should be further encouraged by scientists advising the media on the display of epidemiological data and models. In this context, we also note that data journalists were overall hesitant to use the ensemble nowcasts and prioritized individual nowcasts based on methods described in peer-reviewed publications. Interestingly, the best-performing models in our study were the `MeanEnsemble` and the yet unpublished `ILM` approach. However, our analyses show that all compared approaches provided a good qualitative impression of current incidence trends and levels, and we consider each of them a helpful addition and improvement over showing uncorrected data.

To conclude, we highlight some advantages of the collaborative nowcasting approach adopted in our study. The ensemble nowcast not only showed strong relative performance but was also the most consistently available nowcast, with almost all other models unavailable due to technical problems on some days during the study period. Additionally, our collaborative approach fostered frequent exchange and interaction among modelers via bi-weekly coordination calls, creating a valuable platform for knowledge sharing, feedback, and collaboration on methodological advancements. Through these interactions, the project facilitated model improvements, as seen for the `LMU`, `RKI` and `KIT` approaches in Section 3.6, and fostered discussion on new methodological topics beyond the scope of the present article. For example, the *Epinowcast community* (https://www.epinowcast.org/) was established to build and assess real-time analysis tools, publically available in the R package `epinowcast` [26]. The benefits of our collaborative approach demonstrate the importance of ongoing communication and cooperation in the development and refinement of epidemiological models, particularly during rapidly evolving public health crises such as infectious disease outbreaks.

## Supporting information

**S1 Appendix. Supplementary material.** Additional figures, explanations, and in-depth information.
(PDF)

## Acknowledgments

We would like to thank Tilmann Gneiting for his helpful comments and feedback on the evaluation approach.

## Author Contributions

**Conceptualization:** Daniel Wolffram, Melanie Schienle, Johannes Bracher.

**Data curation:** Daniel Wolffram, Sam Abbott, Matthias an der Heiden, Sebastian Funk, Felix Günther, Davide Hailer, Stefan Heyder, Thomas Hotz, Jan van de Kassteele, Helmut Küchenhoff, Sören Müller-Hansen, Diellë Syliqi, Alexander Ullrich, Maximilian Weigert, Johannes Bracher.

**Formal analysis:** Daniel Wolffram, Sam Abbott, Matthias an der Heiden, Sebastian Funk, Felix Günther, Davide Hailer, Stefan Heyder, Thomas Hotz, Jan van de Kassteele, Helmut Küchenhoff, Sören Müller-Hansen, Diellë Syliqi, Alexander Ullrich, Maximilian Weigert, Johannes Bracher.

**Funding acquisition:** Melanie Schienle, Johannes Bracher.

**Investigation:** Daniel Wolffram.

**Methodology:** Daniel Wolffram, Sam Abbott, Matthias an der Heiden, Sebastian Funk, Felix Günther, Davide Hailer, Stefan Heyder, Thomas Hotz, Jan van de Kassteele, Helmut Küchenhoff, Sören Müller-Hansen, Diellë Syliqi, Alexander Ullrich, Maximilian Weigert, Johannes Bracher.

**Project administration:** Melanie Schienle, Johannes Bracher.

**Resources:** Melanie Schienle.

**Software:** Daniel Wolffram, Sam Abbott, Matthias an der Heiden, Sebastian Funk, Felix Günther, Davide Hailer, Stefan Heyder, Thomas Hotz, Jan van de Kassteele, Helmut Küchenhoff, Sören Müller-Hansen, Diellë Syliqi, Alexander Ullrich, Maximilian Weigert, Johannes Bracher.

**Supervision:** Melanie Schienle, Johannes Bracher.

**Validation:** Daniel Wolffram, Johannes Bracher.

**Visualization:** Daniel Wolffram.

**Writing – original draft:** Daniel Wolffram, Johannes Bracher.

**Writing – review & editing:** Daniel Wolffram, Sam Abbott, Matthias an der Heiden, Sebastian Funk, Felix Günther, Davide Hailer, Stefan Heyder, Thomas Hotz, Jan van de Kassteele, Helmut Küchenhoff, Sören Müller-Hansen, Diellë Syliqi, Alexander Ullrich, Maximilian Weigert, Melanie Schienle, Johannes Bracher.

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
