## [Decision Letter · Decision Letter 0]

12 Jun 2023

Dear Mr. Wolffram,

Thank you very much for submitting your manuscript "Collaborative nowcasting of COVID-19 hospitalization incidences in Germany" for consideration at PLOS Computational Biology. As with all papers reviewed by the journal, your manuscript was reviewed by members of the editorial board and by several independent reviewers. The reviewers appreciated the attention to an important topic. Based on the reviews, we are likely to accept this manuscript for publication, providing that you modify the manuscript according to the review recommendations.

Sincerely,

James M McCaw, PhD

Academic Editor

PLOS Computational Biology

Virginia Pitzer

Section Editor

PLOS Computational Biology

Reviewer's Responses to Questions

**Comments to the Authors:**

Reviewer #1: The paper presents a comprehensive evaluation of real-time nowcasting methodologies as applied to COVID-19 hospitalisation incidence in Germany. The measurement of performance utilised appropriate standard scoring rules, with remarks made on the implications of these results on both public health communication and the statistical design of nowcasting models. In particular, the discussion of what model design decisions likely contributed to under- or over-performance is likely to be valuable for guiding future research. The paper is strengthened through its use of a preregistered study design and reproducible methodology. I note a few suggestions that could be made to improve the paper:

1. Addition of the word COVID-19 to the abstract would make the transition from general discussion of nowcasting to the paper’s specific context clearer.

2. Figures 4 and 5 do not have years present on the x-axis or mentioned in the figure caption. The interpretation of this figure (especially regarding the emergence of Omicron) would be clearer with an explicit year present.

3. The effect of day of week on reporting delays is presented (Figure 2), with some models within the ensemble including this as a covariate (Table 1), however, there is little discussion of the impact of this effect on nowcasting performance. Given the large number of modelling problems where the inclusion of a day of week effect must be considered, reflection on this could be interesting.

Reviewer #2: uploaded as attachment

**Have the authors made all data and (if applicable) computational code underlying the findings in their manuscript fully available?**

Reviewer #1: Yes

Reviewer #2: Yes

PLOS authors have the option to publish the peer review history of their article (what does this mean?). If published, this will include your full peer review and any attached files.

Reviewer #1: **Yes: **Ruarai Tobin

Reviewer #2: No

Figure Files:

Data Requirements:

Reproducibility:

References:

---

## [Editor Report · Decision Letter 1]

28 Jul 2023

Dear Mr. Wolffram,

We are pleased to inform you that your manuscript 'Collaborative nowcasting of COVID-19 hospitalization incidences in Germany' has been provisionally accepted for publication in PLOS Computational Biology.

Best regards,

James M McCaw, PhD

Academic Editor

PLOS Computational Biology

Virginia Pitzer

Section Editor

PLOS Computational Biology

---

## [Editor Report · Acceptance letter]

9 Aug 2023

PCOMPBIOL-D-23-00600R1 

Collaborative nowcasting of COVID-19 hospitalization incidences in Germany

Dear Dr Wolffram,

I am pleased to inform you that your manuscript has been formally accepted for publication in PLOS Computational Biology. Your manuscript is now with our production department and you will be notified of the publication date in due course.

With kind regards,

Zsofia Freund
